# A Review of Graphene-Based Surface Plasmon Resonance and Surface-Enhanced Raman Scattering Biosensors: Current Status and Future Prospects

**DOI:** 10.3390/nano11010216

**Published:** 2021-01-15

**Authors:** Devi Taufiq Nurrohman, Nan-Fu Chiu

**Affiliations:** 1Laboratory of Nano-Photonics and Biosensors, Institute of Electro-Optical Engineering, National Taiwan Normal University, Taipei 11677, Taiwan; devi.taufiq.n@mail.ugm.ac.id; 2Department of Electronics Engineering, State Polytechnic of Cilacap, Cilacap 53211, Indonesia; 3Department of Life Science, National Taiwan Normal University, Taipei 11677, Taiwan

**Keywords:** biosensors, surface plasmon resonance, graphene

## Abstract

The surface plasmon resonance (SPR) biosensor has become a powerful analytical tool for investigating biomolecular interactions. There are several methods to excite surface plasmon, such as coupling with prisms, fiber optics, grating, nanoparticles, etc. The challenge in developing this type of biosensor is to increase its sensitivity. In relation to this, graphene is one of the materials that is widely studied because of its unique properties. In several studies, this material has been proven theoretically and experimentally to increase the sensitivity of SPR. This paper discusses the current development of a graphene-based SPR biosensor for various excitation methods. The discussion begins with a discussion regarding the properties of graphene in general and its use in biosensors. Simulation and experimental results of several excitation methods are presented. Furthermore, the discussion regarding the SPR biosensor is expanded by providing a review regarding graphene-based Surface-Enhanced Raman Scattering (SERS) biosensor to provide an overview of the development of materials in the biosensor in the future.

## 1. Introduction

Graphene, the mother of all carbon materials, has opened a new era in technological development because of its unique properties. Graphene is a single layer of carbon atoms with a 2D hexagonal crystal lattice and is the thinnest and strongest material that has existed to date [1]. Various and unique new properties can be generated by shaping the graphene layer into 0D buckyballs, 1D nanotubes, and 3D graphite [2]. This property makes graphene suitable for applications in various fields, including drug delivery [3], energy storage [4], bioimaging [5], and biosensors [6].

The Surface Plasmon Resonance (SPR) biosensor is a type of biosensor which is very powerful for detecting and determining the specificity, affinity, and kinetic parameters of macromolecular bonds. The working principle of this biosensor uses metals such as gold (Au), silver (Ag), and aluminum (Al) to excite surface plasmon waves which are very sensitive to changes in the refractive index on their surface [7]. Various macromolecular bonds have been detected such as protein–protein, protein–DNA, enzyme–substrate or inhibitor, receptor drug, membrane lipids–proteins, proteins–polysaccharides, and cell—or virus—proteins [8].

The sensitivity of the SPR biosensor is simply defined as the ratio between the shift in wavelength or angle and the change in the refractive index on the sensing surface. This sensitivity is influenced by many factors depending on the excitation method chosen. For the prism coupled SPR biosensor, the sensitivity of the SPR is influenced by the type of prism, metal and wavelength used. Islam et al. reported that for the angle investigation mode, the SPR structure has the best sensitivity with a gold metal with a thickness of 50 nm [9]. For the fiber coupled SPR biosensor with an investigation mode in the form of wavelength, of the four types of metals investigated, the highest sensitivity is gold (Au), silver (Ag), copper (Cu) and aluminum (Al), respectively [10]. For grating coupled SPR biosensor, the sensitivity is influenced by the shape of the grating, the period of the grating, the height of the grating, and the wavelength. Cao et al. reported that in the wavelength range of 600–1000 nm for the angular investigation mode, the best sensitivity belongs to the first diffraction order [11].

The challenge in biosensor development is to detect analytes with small molecular weight and extremely diluted concentrations [12]. At this molecular weight and concentration, conventional biosensors are not able to detect it. Therefore, the development direction of the SPR biosensor is increasing the sensitivity to reach this limit. Several ways have been done and one of the most popular is by exploiting the unique properties of graphene [13]. In this review, the authors summarize the latest developments regarding the graphene-based SPR biosensors. There are several sensing methods discussed in this review, including prism-based, fiber-optic-based, grating-based, nanoparticle-based, and SERS-based SPR biosensors. The discussion was started by presenting a brief theory and continued by presenting the results obtained regarding the graphene-based biosensor, both simulation, and experiment. These results include the sensitivity and detection accuracy obtained due to the presence of graphene in the SPR biosensor.

## 2. Graphene and Its Properties

Graphene is an allotrope of carbon and is intrinsically a zero-gap semiconductor (semimetal). Graphene band structure has a linear energy dispersion and gives rise to two cones crossing at the Dirac point [14]. There are three common forms of graphene, namely graphene oxide (GO), reduced graphene oxide (rGO), pure graphene, and carboxyl-GO (Figure 1). These three forms of graphene exhibit different properties due to their different molecular structural arrangements [15]. GO is composed of a graphene layer with functional groups containing active oxygen on its surface such as carboxyl (–COOH), alkoxy (C–O–C), hydroxyl (–OH), and carbonyl (C–O) groups. The presence of functional groups in GO makes it hydrophilic and highly dispersible in water. However, the functional group breaks the sp^2^ bond in the crystal plane which will make GO non-conductive and has lower mechanical properties than graphene [16]. rGO can be produced by removing the functional groups from the GO and will restore the mechanical and electrical conductivity properties of the graphene layer [17]. The presence of functional groups in rGO makes it quite dispersible in water. Briefly, the physical properties of graphene are shown in Table 1.

Some of the special properties of graphene which make this material very promising to be developed in biosensors in the future, among others, are due to its high electron mobility and high surface to volume ratio. The electron mobility in graphene is ~200,000 cm^2^/Vs. This mobility can still be triggered by applying a gate voltage, making it possible to build graphene-based biosensor with high response speeds. The detection capability of the graphene-based biosensor is still possible to increase to a single molecule by utilizing the surface atoms in the graphene layer [18,19]. Specifically, in the SPR biosensor, the number of graphene layers in plasmonic metal can be controlled so that it is possible to control the SPR response and sensitivity [20]. The presence of graphene can also serve to protect plasmonic metals from oxidation so that their performance is more stable [21,22].

## 3. Graphene-Based Prism Coupled SPR Biosensor

The history of the prism-coupled SPR biosensor dates back to the results of Ritchie’s research in the 1950s. At that time, Ritchie introduced the surface plasmon in detail [23]. From these results, finally Otto in 1968 studied a prism combined with SPR which was based on attenuated total reflection (ATR) [24]. This sensing configuration developed by Otto is known as the Otto configuration and is shown in Figure 2a. In the Otto configuration, the prism and the plasmonic metal are separated by a gap in the order of micrometers. The performance of the SPR biosensor in this configuration is influenced by the air-gap distance between the plasmonic metal and the prism [25]. Due to the difficulty of controlling the gap between the prism and the plasmonic metal, Kretschmann upgraded the Otto configuration as shown in Figure 2b. The plasmonic metal is between the analyte and the prism [26]. Due to the ease of fabrication of SPR structures, the Kretchmann configuration is currently the most common and widely used configuration for SPR sensing applications [27].

The surface plasmon is a quantum plasmonic oscillation produced by the interaction between photons and free electrons possessed by metals. Resonance occurs when the surface plasmon (SP) wave vector matches the incident light wave vector [1,27]. Theoretically, the SP wave vector can be obtained from Maxwell’s theory. Applying the appropriate boundary conditions, the wave vectors of the surface plasmon are:(1)$$K_{sp}=\frac{\omega }{c}\sqrt{\frac{\varepsilon_{m}\varepsilon_{d}}{\varepsilon_{m}+\varepsilon_{d}}}$$
and the wave vector of incident light is:(2)$$K_{x}=\frac{\omega }{c}\sqrt{\varepsilon_{p}}\sin\theta_{i}$$
where $\;\varepsilon_{p}$, $\varepsilon_{m}$, and $\varepsilon_{d}$ indicate the dielectric constant of the prism, metal, and dielectric, respectively, while $\theta_{i}$ indicates the incident angle. Resonance can be achieved by changing the incidence angle and it can be predicted mathematically by the following equation [28]:(3)$$\theta_{res}=\sin^{-1}\left(\frac{1}{\sqrt{\varepsilon_{p}}}\sqrt{\frac{\varepsilon_{m}\varepsilon_{d}}{\varepsilon_{m}+\varepsilon_{d}}}\right)$$
where $\theta_{res}$ indicates the resonance angle and in some literature it is called the SPR angle. The dispersion relation of the SPR biosensor coupled with the prism is shown in Figure 2. It can be seen that by using the refractive index material $n_{p}$, the incident light propagation constant k is capable of coupling the SP wave vector at the intersection point representing the resonance conditions.

A number of studies, both simulations and experiments, have been carried out to obtain a good performance SPR biosensor. In a simulation study, the Fresnel equation and the transfer matrix method (TMM) can be used to determine reflectance in multilayer SPR structures [29]. From the resulting SPR spectrum, the performance of the SPR biosensor coupled with a prism can be evaluated through the value of sensitivity (S), detection accuracy (DA), and quality factor (QF). These three values must be as high as possible to get the best performing biosensor. Sensitivity (S), detection accuracy (DA), and quality factor (QF) can be determined by the following equation:(4)$$S=\frac{\Delta \theta_{SPR}}{\Delta n_{Bio}}$$
(5)$$DA=\frac{\Delta \theta_{SPR}}{\mathrm{FWHM}}$$
(6)$$QF=\frac{S}{\mathrm{FWHM}}$$

In the above equation, $\Delta \theta_{SPR}$ shows the shift in the SPR angle, $\Delta n_{Bio}$ shows the change in the refractive index of the sensing medium, and FWHM shows the full width at half maximum [30].

The effect of adding a graphene layer to the prism-based SPR on its performance was described by Wu et al. in 2010 and Choi et al. in 2011. Wu et al. investigated the effect of adding a graphene layer to the gold surface (Figure 3a). At the same refractive index change ($\Delta n_{Bio}=$ 0.005), the SPR angle shifts in conventional biosensors and graphene monolayer-based biosensors were 0.26 and 0.266, respectively. The results of calculations by Wu et al. also show that the graphene on gold SPR biosensor with a graphene (L) layer is (1 + 0.025 L) × γ (where γ > 1) times more sensitive than conventional gold SPR biosensors. However, adding more layers of graphene to the gold will widen the SPR curve. This will result in difficult reading of the SPR angle during the experiment so that there will be potential for reading errors [31].

Choi et al. investigated the effect of the graphene layer on a different metal, namely silver (Figure 3b). Increasing the number of graphene layers results in a higher reflectance at the SPR angle and a shallower SPR curve. In addition, a decrease in sensitivity due to an increase in the number of graphene layers cannot be avoided. However, it should be noted that the presence of a graphene monolayer and bilayer in the SPR biosensor results in a remarkable increase in SPR sensitivity. In this structure, the SPR sensitivity is 3.5 and 2.5 times higher than that of conventional gold-based SPR biosensors [32].

The results of Wu et al. and Choi et al.’s research inspired other researchers in the field of SPR biosensors. Maharana et al. compared chalcogenide and silicon prisms in an SPR biosensor composed of gold and a graphene monolayer (Figure 4a). The results obtained show that the biosensor accuracy is up to 100% compared to the silica-based SPR biosensor [33]. In the same year, Verma et al. added a material with a high refractive index, namely silicon (Si), to the graphene-based SPR structure (Figure 4b). The results obtained show that the best performance is owned by the SPR biosensor with gold (40 nm)/Si (7 nm)/bilayer graphene. In this structure, the shift in the SPR angle is more than twice that reported by Wu et al. [34]. In 2014, Ryu et al. also investigated the effect of GO on SPR sensitivity through simulations and experiments on structures composed of gold, GO, and SiO_2_ as a spacer layer between gold and GO. The results obtained show that the SPR biosensor coupled with GO has an SPR angle shift of 13% than the conventional structure [35].

Another method that can be used to improve the performance of the SPR biosensor is long-range surface plasmon resonance (LRSPR). LRSPR is a development of conventional SPR biosensor by adding a dielectric buffer layer (DBL) with a low refractive index such as Fluoride, Cytop, and Teflon. When applied for sensing purposes, LRSPR will exhibit a narrower and deeper resonance dip and allow it to penetrate deeper into the analyte due to less loss than conventional SPR biosensor. There are two types of LRSPR structures, namely the common LRSPR (DBL is between the prism and the metal) and the symmetric LRSPR (the metal is sandwiched between two DBLs) [36].

Wu et al. investigated LRSPR on graphene in 2016. They investigated three different metals (Al, Ag, and Cu) with Cytop as their DBL. The structure of the SPR investigated was common LRSPR, which was composed of 2S2G prism/Cytop/metal/graphene/analyte. Figure 5 is the SPR spectrum for the standard SPR and LRSPR structures. Figure 5a shows the spectrum in Al metal, Figure 5b shows the spectrum in Cu metal, and Figure 5c shows the spectrum in Ag metal. In the three different metals, the LRSPR spectrum shows a shape with a very small FWHM compared to the standard structure. The FWHM of the LRSPR and SPR biosensors are 0.0115° and 0.1408° for Al-graphene based configurations, 0.0099° and 0.1869° for Cu-graphene based configurations, 0.0123° and 0.3854°, respectively, for Ag-graphene based configurations. This shows that LRSPR has succeeded in reducing FWHM and enhancing the accuracy of the SPR biosensor [37].

Apart from simulation studies, experimental studies on graphene for SPR biosensors have also shown very fast progress and development. In a group led by prof. Chiu, the experiment was started in 2012. The research was started by investigating the performance of the SPR biosensor integrated with loop-mediated isothermal amplification (LAMP) on single-layer GO and rGO with cystamine (Cys) as a linker to detect tuberculosis bacterial DNA (TB DNA) (Figure 6a). To determine the resulting performance, three different sensing surfaces were fabricated, namely Cys-linker, Cys-GO, and Cys-rGO. Figure 6b shows the SPR response after the TB DNA was injected into each sensing surface. If we look at the SPR response, two important results are obtained from this experiment. First, the Cys-GO sensing layer has the best sensitivity because it has a higher SPR angle shift. Second, the Cys-GO sensing layer has excellent stability. After the sensing layer was regenerated with NaOH, the baseline did not decrease. The presence of the -COOH group on GO results in a very strong covalent bond between the developed substrate and TB DNA [38]. The effect of the number of GO and rGO layers on SPR sensitivity was also investigated by Chung et al. GO with different number of layers was fabricated by alternative dipping of gold substrate in positive and negatively charged GO solutions. Meanwhile, the rGO layer was obtained by reducing GO using hydrazine. From several structures investigated, the three-layer GO based chip showed the best performance. The resulting SPR sensitivity for this structure was 150.38 °/RIU which was 3.45% higher than the bare gold substrate [39].

Chiu et al. (2014) used a previously studied to study the interaction between antibody (BSA) and antigen (anti-BSA). Figure 7a shows the biomolecular bonding mechanism between BSA and anti-BSA and Figure 7b shows the response of the SPR biosensor. At all detected anti-BSA concentrations (75.75 nM, 151.51 nM, and 378.78 nM), the GO-based SPR biosensor showed a higher response. Specifically, at the anti-BSA concentration of 75.75 nM, the SPR angle shift in the GO-based SPR structure was 1.4 times higher than that of the conventional structure; whereas at the highest concentration (378.78 nM), the SPR angle shift was up to two times higher than that of conventional structure. These results indicate that the GO-based SPR structure has a better performance than conventional structures [40].

Chiu et al. also conducted an experimental study of other structures, namely GO-based structures that functionalized with –COOH (GO–COOH chip) as an immunosensor to detect non-small cell lung carcinoma via cytolerayin 19 (CK19) (Figure 8a). Biomolecular interactions between CK19 and anti CK19 at different concentrations were measured in real time to obtain sensorgram data for each fabricated chip. Based on sensorgram data, the GO–COOH-based SPR chip has a shorter response time than conventional chip. In addition, functionalization of –COOH on GO resulted in a better detection limit of 0.001–100 pg/mL (Figure 8b).

The use of graphene for biomolecule detection has also been carried out by other research groups. He et al. used a fabrication method for the growth of graphene layers on cooper foil developed by Li et al. to construct an SPR biosensor for the purpose of detecting serum folate biomarkers [42]. The specific recognition of FAP is based on the interaction between the integrated folic acid receptors via the buildup of π on the graphene-coated SPR chip and the FAP analyte in serum. To block non-specific interactions on the SPR chip, human serum (HS) is mixed with bovine serum albumin (BSA). Figure 9a shows the SPR response at different FPA concentrations ranging from 10 fM to 1 pM. Based on this SPR response, a linear range was obtained up to a concentration of 500 fM. In this range, the correlation coefficient R^2^ is 0.999 and the relationship between the FPA concentration and the SPR response can be determined as 1.62 + 25.85 × [FAP]. In addition, the manufactured chips also have excellent reproducibility. Over the twenty-day time span, there was no significant change in the SPR signal. A sensor interface with a long lifetime was obtained in this experiment (Figure 9b) [43].

Omar et al. developed an SPR biosensor based on the rGO–Polyamidoamine (rGO-PAMAM) composite that functioned with amines to detect and measure the dengue virus. Gold film is inserted into succinimidyl undecanoate (DSU) solution to form a self-assamble monolayer (SAM) which functions for chemisorption of biomolecules through amide relations. After SAM was formed, rGO-PAMAM was deposited on the gold surface using the spin coating method and followed by immobilization of specific antibodies for DENV 2 E-Protein via EDC/NHS using the same method. From this experimental scheme, the SPR chip formed is called the Au/DSU/rGO-PAMAM/Ab chip.

Figure 10 shows the calibration curve and selectivity of a fabricated SPR chip. The calibration curve was obtained based on the shift in the SPR angle at different concentrations of DENV 2 E-Proteins from 0.08 pM to 1 pM. The linear range is obtained in this concentration range with the correlation coefficient R^2^ is 0.92577. The selectivity of the SPR chip was also tested by detecting other types of antigens, namely DENV E-proteins and ZIKV E-proteins. Based on Figure 10b, the shift of SPR angle in a solution of DENV E-proteins and ZIKV E-proteins is very much smaller when compared to DENV 2 E-Proteins; whereas in both antigens the sample concentration was 10 pM and in the DENV 2 E-Proteins solution was 0.1 pM. From this experiment, it can be concluded that the sensitivity and selectivity of the sensor are very good with a detection limit of 0.08 pM [44].

The prism-coupled SPR biosensor is a popular platform and is used in many detection cases. The preceding discussion describes several approaches that have been investigated to improve the performance of SPR biosensor. Table 2 below shows the resulting performance of the various types and structures of analytes.

## 4. Graphene-Based Fiber Coupled SPR Biosensor

The propagation of light in optical fiber also works based on total internal reflection (TIR). The prism in the prism-based SPR biosensor functions to produce TIR on the metal–prism surface. For this reason, the prism can be replaced with a core from an optical fiber to form a fiber-optic-based SPR biosensor [55]. The sensing surface of an optical fiber can be formed by removing a small portion of the fiber to be coated with a thin layer of metal. A polychromatic light source is launched from one end of the optical fiber and the other end measures the transmitted light [56]. Resonances occur at specific wavelengths and biomolecular interactions can be identified by shifting the resonant wavelengths. Several parameters that affect the performance of the SPR biosensor for this platform are the length of the sensing region and the diameter of the fiber core.

Simulation of the effect of graphene on conventional optical fiber SPR biosensors carried out by Fu et al. in 2012. Fu et al. investigated the sensitivity changes in fiber clading with sensing medium N = 5 mm, gold thickness 40 nm, number of graphene layers N = 5, diameter of fiber cores. D = 50 micrometers, and the refractive index of fiber core and clading are 1.451 and 1.45, respectively. The complete experimental scheme is shown in Figure 11a. Figure 11b shows the SPR spectrum of the simulation results which shows the relationship between the sensing surface refractive index and the SPR wavelength shift. At three different refractive indexes (1.33, 1.35, and 1.37), the SPR biosensor with graphene showed a higher wavelength shift than without graphene. This suggests that the presence of graphene in fiber-optic-based biosensors can also increase SPR sensitivity [57].

Another research group, Zhou et al. investigated the effect of single layer graphene on SPR sensitivity in end reflection optical fibers with a coroless fiber diameter of 600 micrometers and a silver thickness of 40 nm (Figure 12a). The results showed that in the refractive index range from 1.3411 to 1.3737, the SPR sensitivity without and with graphene was 2657 nm/RIU and 3091 nm/RIU, respectively [58]. The simulation results reaffirm that graphene in conventional optical fibers is proven to increase sensor sensitivity.

Zhuo et al. have also succeeded in confirming the simulation results obtained through experiments. The sensing part (about 10 mm) of the probe was immersed in acetone to soften the clading layer so that the clading layer on the sensing part could be removed easily. After that, the sensing part was coated with Ag film by chemical method and followed by transferring graphene on the optical fiber. Briefly, the transferring process begins by immersing the PMMA/graphene/Cu film in ferric chloride (FeCl_3_) to etch the Cu foil, placing the PMMA/graphene film on the sensing surface, heating the graphene and removing PMMA by immersing the sensing probe in acetone. The fabricated probes are then tested by detecting NaCl with different concentrations. At the same concentration change, the shift in the SPR wavelength for graphene-based optical fibers shows a higher shift (Figure 13). Based on the fitting curve data, the sensitivity for probes without and with graphene were 4.05 nm/% (2487.7 nm/RIU) and 6.417 nm/% (3936.8 nm/RIU), respectively [58].

Another optical fiber type for SPR biosensor applications is photonic crystal fiber (PCF). Li et al. investigated an H-formed photonic crystal fiber SPR biosensor with U-shaped grooves open structure for refractive index sensing. PCF biosensors are composed of two layers of air holes in a hexagonal layout. The center air outlet can lower the effective RI from core guiding mode to in phase with plasmon mode. There are two large air holes in the first layer, leading to the phenomenon of strong birefringence and a highly polarized light connection with the metal dielectric interface. In this investigation, the designed structure has a gap between two air holes Λ = 2 µm, d_c_/ Λ = 0.5, d_1_/ Λ = 0.5, d_2_/ Λ = 0.9, d_3_/ Λ = 0.8, t_Ag_ = 40 nm, and t_graphene_ = 4.08 nm, where d_c_, d_1_, d_2_, d_3_ are the diameter of the air holes in PCF, and the t_Ag_ and t_graphene_ show the thickness of the silver and graphene layers, respectively. In simple terms, the schematic of the investigated PCF biosensor is shown in Figure 14a. Figure 14b shows the biosensor response at different refractive indications from 1.33 to 1.41. Based on the results of the fitting curve, in this refractive index range there are two slopes which indicate the sensitivity of the biosensor. The first sensitivity was 2770 nm/RIU in the refractive index range from 1.33 to 1.36, and the second sensitivity was 6057 nm/RIU in the refractive index range from 1.36 to 1.41 [59].

The last configuration is optical fiber with a nano-structured coating to produce the localized surface plasmon. One of the studies using this configuration was carried out by Huang et al. in 2019. Huang et al. used a multimode fiber-single mode–multimode (MMF–SMF–MMF) structure coated with a graphene-metal hybrid for temperature sensor. The sensor sensitivity was further enhanced by the addition of core-shell gold–silver nanoparticles (Au@AgNPs) modified with polydimethylsiloxane (PDMS) as a material for temperature sensing. PDMS is a material with a high thermo-optical coefficient (−4.5 × 10^−4^/°C) and its refractive index decreases with increasing temperature [60]. Briefly the probe fabrication process is shown in Figure 15a. Figure 15b shows the biosensor response at different temperatures. As the temperature increases, the SPR wavelength shifts to a smaller wavelength. The SPR wavelength shifted from 871.41 nm to 798.60 nm when the temperature changed from 30 °C to 110 °C. The repeatability of the biosensor was also tested through heating and cooling processes. Based on the fitting curve in Figure 14c, the correlation coefficient R^2^ is 0.993 which indicates the biosensor has good repeatability. From the slope obtained from the fitting process, the sensitivity of the MMF–SMF–MMF structure with graphene-gold-Au@Ag NPs-PDMS is −1.02 nm/°C [61]. The sensitivity of this structure is higher than PDMS-long period fiber gratings coated with PDMS (0.2554 nm/°C) [62], graphene quantum dots-coated hollow core fiber (0.1237 nm/°C) [63], and Multimode Fiber—Fiber Bragg Grating—Multimode Fiber (MMF-FBG-MMF) structure (0.172 nm/°C) [64].

The previous discussion describes four configurations in the SPR fiber-optic-based biosensor and papers relating to the use of graphene in each configuration. Table 3 below shows a summary of research results related to the SPR biosensor which is coupled with a graphene-based optical fiber.

## 5. Graphene Based Grating Coupled SPR Biosensor

The SPR biosensor coupled with a grating as shown in Figure 16 has been an important pioneer in the SPR configuration. This grating configuration was first introduced by Wood in 1902 [70]. To obtain plasmon and photon resonance, both the wave vector and the effective refractive index of the guided wave must meet the following equation [71,72]:(7)$$k_{sp}^{\left(m\right)}=k_{x,photon}\sin\theta_{i}\pm m\;k_{grating}=\frac{2\pi }{\lambda }n_{p}\sin\theta_{i}\pm m\frac{2\pi }{P}$$
and
(8)$$n_{eff}=n_{b}\sin\theta_{res}\pm m\frac{\lambda }{P}$$

In the above equation, $k_{x,photon}$, $k_{grating}$, and $k_{sp}$ show the wave vector of incident light, grating, and surface plasmon, respectively. Next, $n_{b}$, $m_{th}$, and P show the refractive index of the medium, the diffraction order and the grating period, respectively. For the sub wavelength grating-mediated interactions between surface plasmon and analytes, the following momentum matching relation is preserved:(9)$$k_{sp}^{\left(m\right)}=\frac{2\pi }{\lambda }n_{p}\sin\theta_{res}\pm m\frac{2\pi }{P}=\frac{2\pi }{\lambda }\sqrt{\frac{\varepsilon_{m}\varepsilon_{D,eff}}{\varepsilon_{m}+\varepsilon_{D,eff}}}$$

The performance of the SPR-based grating biosensor is influenced by several factors, including the shape of the grating, the operating wavelength, the period of the grating, the type of metal, and the refractive index of the analyte [71]. Sadeghi and Shirani investigated the performance of the graphene–gold ellipse grating SPR biosensor in the mid-infrared region. The schematic diagram of the investigated biosensor is shown in Figure 17a. The initial parameters of the SPR biosensor are as follows: grating period (p) = 1500 nm, perpendicular radius (T_1_) = 56 nm, gold thickness (h) = 10 nm, horizontal diameter (T_2_) = 46 nm, and the height of the sensing medium (Ds) = 600 nm. Based on the profile of the extinction curve at wavelengths from 1500 nm to 2500 nm, the peak extinction is at a wavelength of 2001.2 nm. Figure 17b shows the electromagnetic distribution profile at this wavelength. The intensity of the magnetic field increases exponentially with the closer to the interface in the metal medium. However, the magnetic field decreases slowly in the dielectric medium. Furthermore, the sensitivity of the biosensor is obtained from the shift in peak extinction wavelength to changes in a certain refractive index ($S=\Delta \lambda_{peak}/\Delta n$). In the refractive index change Δn = 0.001 from 1.333 to 1.334, the resulting SPR biosensor sensitivity is 1450 nm/RIU. After the optimization process, the best SPR sensitivity generated in this structure is 1782 nm/RIU [74]. Sadeghi and Shirani also investigated the performance of the SPR biosensor on rectangular gratings. The resulting sensitivity for this structure is 1180 nm/RIU [75].

In the mid infrared range, noble metal exhibits large ohmic losses due to its low charge-carrier mobility and large permittivity [76]. Wei et al. simulated a conformal graphene-decorated nanofluidic channel (CGDNC) based on surface plasmons at infrared frequencies. The CGDNC schematic and transmittance spectrum at different refractive indices are shown in Figure 18 below. The sensitivity at different period ∧, width W, height H, and Fermi level $E_{f}$ is determined by the equation $S=\delta \lambda_{GSP}/\;\delta n_{d}$. The results obtained show that at a period of 100 nm to 300 nm and a height of 20 nm to 100 nm, the best sensitivity is owned by the structure with the highest period and height, namely 300 nm and 100 nm. The sensor sensitivities in these dimensions are 4356 nm/RIU and 3693 nm/RIU, respectively. On the other hand, if the sensitivity is investigated at different Fermi width and energy, the best sensitivity is the structure with the smallest fermi width and energy. At a width of 20 nm and a fermi energy of 0.1 eV, the sensitivity obtained was 6050 nm/RIU and 8004 nm/RIU, respectively [77]. The most important result of this simulation shows that the sensor sensitivity can be actively increased by lowering the Fermi energy level of graphene which can be done by adjusting the external voltage [77,78].

In 2013, Reckinger et al. fabricated holey gold films with colloid nanosphere lithography with grating parameter of 980 nm and holes diameter of 405 nm for ethanol detection. The fabricated film was then confirmed based on a Scanning Electron Microscope image (Figure 19a). Graphene is then grown on the gold surface to improve the biosensor’s performance. To determine the resulting performance, ethanol is exposed to the sensor interface coated and without graphene. The ethanol level detected was determined based on the transmission peak shift as shown in Figure 19b. The transmission peaks shifted by 105 from 1460 nm to 1565 nm for bare metal and by 140 nm from 1475 nm to 1615 nm for graphene-coated gold. The sensitivity of the graphene-coated sensor is improved with a transmission peak shift 33% higher than that of bare gold [79].

Wei et al. investigated the effect of graphene on LPG to detect methane. Long period fiber grating (LPFG) with a grating period of 600 micrometers was fabricated by the writing method of high frequency CO_2_ laser pulses. The dimensions of the sensing medium (L), fiber core diameter (D), and metal thickness (T) are shown in Figure 20a. Figure 20b shows the relationship between methane concentration and resonance wavelength shift in three different structures, namely LPFG, LPFG + Au, and LPFG + Au + graphene. In the methane concentration range below 3.5%, the three structures have good linearity with the correlation coefficients R^2^ of LPFG, LPFG + Ag, LPFG + Ag + graphene being 0.975, 0.995, and 0.995. The sensitivity and detection limits obtained were 0.116 nm /% and 0.086% for LPFG, 0.262 nm /% and 0.038% for LPFG + Ag, and 0.34 nm /% and 0.029% for LPFG. The presence of graphene on the silver surface of LPFG can increase sensor sensitivity by 1.31 times that of bare silver LPFG [80].

Although SPR biosensors coupled with gratings are pioneers in the field of SPR biosensors, experimental studies related to the use of graphene for this platform are still rare. Table 4 below shows some of the applications of this platform and the resulting sensitivity.

## 6. Graphene-Based Nanoparticle Coupled SPR Biosensor

Another type of SPR biosensor is the SPR biosensor which uses nanoparticles as a signal amplifier. Of the various types of nanoparticles, plasmonic nanoparticles are the most preferred because of their properties that can produce localized surface plasmon resonance (LSPR) which can increase local electromagnetic fields [82] and increase SPR response [83]. Gold nanoparticles (AuNPs) are recognized as the best material for immunoassay because of their extraordinary properties such as easy reductive preparation, exceptional optical property, water solubility, and significant bicompability. For silver nanoparticles (Ag NPs), this material produces a sharper peak and can be used to increase the sensitivity of the biosensor.

Taking advantage of the unique properties of gold, silver and graphene, Zhang et al. decorated gold or silver nanoparticles with graphene to detect mouse IgG. To be able to capture the IgG mouse, the hybrid nanoparticles are functionalized with goat anti human IgG. Next, solutions containing different concentrations of target human IgG are flushed onto the sensing surface to evaluate the performance and detection limits of the system. Of the three fabricated systems, Ag-graphene hybrid nanoparticles showed the best response compared to Au-graphene hybrid nanoparticles and unmodified biosensors (Figure 21a). The detection limits of Ag-Graphene and Au-Graphene are 0.15 µg/mL and 0.30 µg/mL, respectively. The selectivity of the biosensor was also tested by detecting human IgG and bovine IgG. There is no observable shift in the resonant wavelength (Figure 21b). This demonstrates the selective binding of the system that was created with the IgG mouse [84].

Chiu et al. proposed a colorimetric LSPR immunoassay based on the AuNPs-GO hybrid to detect disease biomarkers and rapidly diagnose infectious diseases. The binding of AuNPs-GO- anti-BSA to GO-BSA is detected by monitoring changes in optical absorbance at a specific wavelength. Figure 22a shows the absorbance spectra at different anti-BSA concentrations from 145 fM to 1.45 nM. There are two absorbance peaks at the wavelengths of 540 nm and 760 nm. From this absorbance peak, a calibration curve is obtained which states the relationship between the anti-BSA concentration and the shift in the absorbance peak at the two peaks (Figure 22b). The detection limit for this sensor is 145 fM with a linear range from 145 fM to 1.45 nM [85].

The same structure was also applied to gas detection by Cittadini et al. They developed gas sensor based on LSPR using rGO coupled with a gold monolayer. Gold monolayer was fabricated by spinning gold colloids with a concentration of 15 and 30 mM at a speed of 3000 rpm for 30 s. The gold monolayer produced from gold colloids for a concentration of 15 mM was indicated by L (AuL) and for a concentration of 30 mM indicated by H (AuH). The two monolayers of gold are then coated with rGO by the spin method and produce a sensing layer called AuL-rGO and AuH-rGO. The sensing layer was then tested by exposing different gas types, namely H_2_ (10,000 ppm and 100 ppm), CO (10,000 ppm), and NO_2_ (1 ppm). The results obtained indicate that the sensor shows a good and reversible response with fast kinetics to H_2_ and NO_2_, while no response was detected to CO [86]. A summary of the SPR biosensor coupled nanoparticles based-graphene is shown in Table 5.

## 7. Graphene-Based Plasmon Coupled Emission Biosensor

In 2003, Lakowicz et al. developed the SPR biosensor with a fluorescence technique called surface plasmon coupled emission (SPCE) [90]. The working principle of this biosensor is based on the fluorescence and plasmonic properties of the nanostructures. Fluorescence molecules are placed on a metal surface with a thickness of 20–50 nm [91]. Next, a light source of a specific wavelength is directed through the prism (Figure 23a) or from the sample side (Figure 23b) to excite the SP wave and high directional emission [91,92]. When the analyte binds to the fluorescence molecule, an energy shift occurs in the fluorescence spectrum. This shift is the basis for the application of SPCE for biosensor applications.

Mulpur et al. engineered and investigated the SPCE properties of the graphene–silver thin film stack as a function of the graphene thickness. Different types of graphene, both single layer and multilayer, are fabricated by the CVD method compared to graphene which is fabricated by the chemical exfoliation method. The resulting graphene is then grown on a silver surface using the spin coating method. Flourescence measurements were carried out at a wavelength of 532 nm with a reverse Kretschmann (RK) configuration (Figure 24a). Figure 24b shows the fluorescence intensity in different structures where the exfoliated graphene-based substrate has the highest intensity. The fluorescence intensity at EG is forty times that of the free space structure [94].

Nanocubes (NCs) have a special edge structure and tip effect that can induce strong localization and increased localized surface plasmons resonance fields. Therefore, Xie et al. performed an enhanced fluorescence signal using silver nanocubes (Ag NCs) and GO on a gold film. The SPCE signal is measured in the reverse kretschmann (RK) configuration at a wavelength of 532 nm (Figure 25a). The results obtained showed that the fluorescence intensity of the SPCE structure with AgNCs and GO had a signal intensity thirty times higher than the structures without GO and NCs (Figure 25b) [95].

Xie et al. amplified the SPCE signal in gold using GO and applied it to detect human IgG. The fluorescence signal from SPCE was measured with a Reverse Kretschmann (RK) configuration at a wavelength of 532 nm. The results obtained indicate that GO can increase the fluorescence intensity up to seven times that of the SPCE structure without GO (SPCE (Au)) or twenty-five times that of the free space emission (FSE). Furthermore, different human IgG concentrations were measured by SPCE (Au + GO) and SPCE (Au) and the fluorescence intensity at each concentration was then measured. The results obtained indicate that the linear relationship is shown in SPCE (Au + GO) with a wider range compared to SPCE (Au) (Figure 26). At semilogarithmic coordinates, the linear relationship and correlation coefficient R^2^ are 0.01–800 ng/mL and 0.993 for SPCE (Au + GO) and 1–100 ng/mL and 0.926 for SPCE (Au), respectively. The detection limits for SPCE (Au + GO) and SPCE (Au) were 0.006 ng/mL and 0.15 ng/mL [96].

The previous discussion describes the working principle of SPCE and the use of graphene for this platform of biosensor. Until now, graphene has not been widely applied to this platform. Table 6 below is a summary of the results of research related to graphene-based SPCE.

## 8. Graphene-Based SERS Biosensor

Raman spectroscopy is a vibration spectroscopy technique that measures the inelastic scattering of light. The resulting Raman spectrum provides a specific fingerprint regarding the molecular structure and material composition [99]. In biosensors, this fingerprint can be used to directly identify an analyte and determine its level [100]. However, Raman scattering is very weak and will not be seen especially when identifying organic molecules which have high fluorescence. The presence of fluorescence can inhibit the identification of molecules especially at very low concentration levels [101,102].

The latest development in analyte detection which utilizes Raman signals is the surface-enhanced Raman scattering (SERS) technique. This technique, first introduced by Fleischmann, Hendra, and McQuillan in 1973 [103], was accomplished by adding metal nanostructures to increase the weak signal and reduce the risk of fluorescence interference as illustrated in Figure 27 [104]. The increase in the Raman signal in this technique occurs due to SPR, which is when the laser excitation energy is close the surface plasmon energy of the metal substrate. This technique is called electromagnetic enhancement [105]. There is another technique in which the Raman signal does not depend on the substrate but on the molecular analyte. By taking advantage of the increased probability of the Raman transition when the analyte is absorbed, this technique is known as chemical enhancement [104,105,106].

Graphene has been shown to be a very outstanding SERS active substrate due to its advantages over other traditional materials. By utilizing its large surface area and superior adsorption capabilities, graphene can be utilized to extinguish photoluminescence of fluorescent dyes and drastically eliminate the fluorescence background [108]. Xie et al. developed a graphene substrate grown on SiO_2_/Si surfaces to suppress photoluminescence. Graphene is fabricated by mechanical oxfoliation from graphite. Figure 28b shows the Raman spectrum of rhodamine 6G (R6G) in aqueous solution (10 µM) and R6G in single layer graphene. From the two spectra obtained, the graphene substrate showed a clearer Raman signal without fluorescence background than the R6G solution. The photoluminescence suppression effect is due to electron transfer and energy transfer between the graphene and the R6G dye molecule [109]. In 2010, Ling et al. compared SERS substrate based on graphene and SiO_2_/Si. To determine the performance of the two substrates, R6G was deposited on the two substrates using a solution-soaking method. As shown in Figure 28c, after solution-soaking, the Raman signal intensity of R6G on single layer graphene was much stronger than on SiO_2_/Si substrate. In addition, the Raman signal in single layer graphene can still be detected clearly even though the R6G concentration is as low as 1 nM [110].

Graphene only supports chemical enhancement and does not support electromagnetic enhancement due to its smooth surface and high optical transmission over the visible range [111]. To overcome this shortcoming, one approach to obtain high performance SERS substrates with chemical and electromagnetic enhancement is by fabricating the substrate which is composed of graphene and metal nanoparticles [112]. Lin et al. studied DNA hybridization by utilizing self-assembly of Ag NPs and SERS substrate based on Ag NPs and graphene (Ag NPs–graphene) nanocomposites. AgNPs are modified with non-fluorescent 4-mercaptobenzoic acid (4-MBA) which is a very efficient Raman probe for DNA hybridization. Furthermore, automatic assembly can occur by functionalization of Ag NPs-GO with DNA probes and Ag NPs with DNA targets. The experimental schematic is shown in Figure 29a. As shown in Figure 29b, the detection limit of the SERS-based DNA sensor was determined by measuring the SERS intensity with a variation of the target DNA concentration up to a concentration of 1 pM. The characteristic peak at 1078 cm^−1^ can be clearly observed. More importantly, the SERS intensity at 1078 cm^−1^ increases linearly with the logarithm of the target DNA concentration in the range 10^−6^–10^−12^ M. The detection limit obtained from this experiment is 10^−14^ M [113]. In addition to DNA hybridization applications, the SERS substrate based on Ag and graphene nanocomposites has also been utilized for sensing inorganic ions, dye molecules, and pesticides [114].

The second metal type that is widely used as a SERS substrate is gold. One of the advantages of this material is its higher stability than Ag NP. Possible interactions between graphene and gold can occur in several ways including covalent bonds when graphene, gold, or both are functionalized [115], non-covalent attachments in the form of π–π interactions [116], and van der walls forces which may occur in unmodified graphene [117]. Until now, the SERS substrate based on graphene and gold has been successfully fabricated using self-assembly [116], electrochemical deposition [118], or in situ growth processes [119] and has employed for biosensing cancer and cancer steam cells [120], multiplex DNA detection [121], or as Hg^2+^ sensors [122,123]. Table 7 below shows a summary of some of the literature related to graphene-based SERS biosensors.

## 9. Conclusions

Graphene and its derivatives have proven to be one of the best materials to enhance the performance of SPR biosensors in many platforms. The simulation study of the graphene-based SPR biosensor has been proven experimentally. To date, experimental studies have reported that the graphene-based SPR biosensor is capable of detecting analytes up to nM or ng/mL levels. Some researchers have also claimed that they were able to detect analyte to smaller levels (fM or fg/mL and aM or ag/mL). This shows that graphene is a superior material and is very promising in the development of SPR biosensors in the future.

One of the challenges in developing graphene-based biosensors is to reduce noise in the detection process of very complex human samples such as in the form serum, plasma, urine, and stool. Modification of the sensing surface is needed to reduce noise so that the detection process is very selective and accurate. For the commercialization of graphene-based SPR biosensors, a simple SPR chip fabrication method, which can control the chip dimensions precisely and accurately in large quantities, is required. This is very important to ensure the resulting performance is as desired.

## Figures and Tables

**Figure 1 nanomaterials-11-00216-f001:**
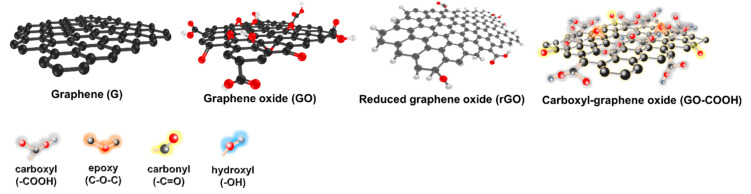
Graphene and its derivatives.

**Figure 2 nanomaterials-11-00216-f002:**
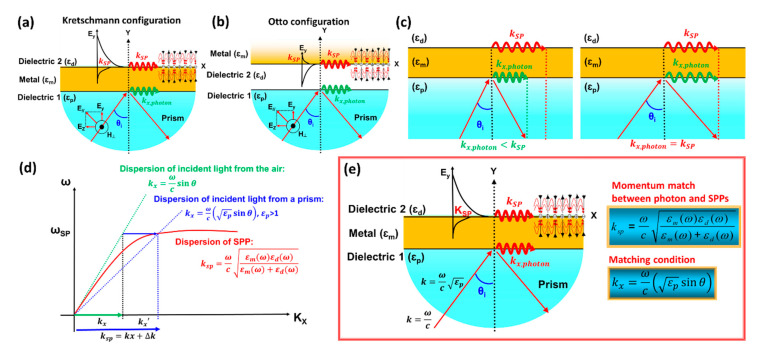
Configurations of the attenuated total reflection (ATR) method: (**a**) Otto configuration and (**b**) Kretschmann configuration. (**c**) The dispersion curve for a surface plasmon mode shows the momentum mismatch problem between the free space photon (k_photon_, gree line) and surface plasmon modes (k_SP_, red line). (**d**) Dispersion relation of surface plasmon with photon. (**e**) The same effect of momentum-supply can be achieved by corrugating the metallic surface in the so-called prism-coupled SPR. The resonance is by using evanescent wave produced in attenuated total reflection (ATR).

**Figure 3 nanomaterials-11-00216-f003:**
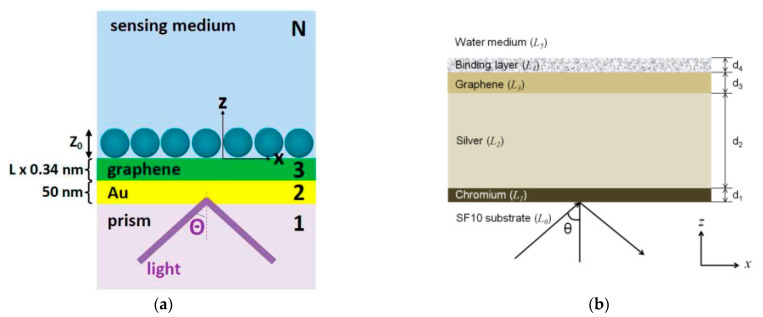
The SPR structure investigated by: (**a**) Wu et al. (left) and (**b**) Choi et al. (right) (adapted with permission from References [31,32] © The Optical Society).

**Figure 4 nanomaterials-11-00216-f004:**
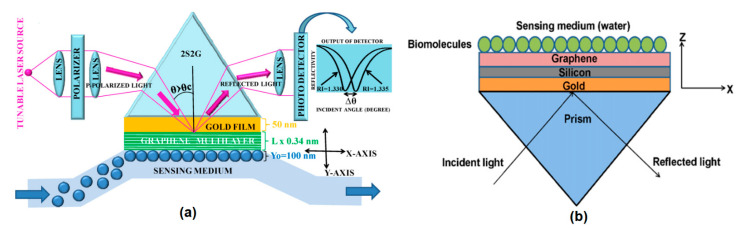
The SPR structure investigated by (**a**) Maharana et al. (left) and (**b**) Verma et al. (right) (adapted with permission from References [33,34], copyright Elsevier).

**Figure 5 nanomaterials-11-00216-f005:**
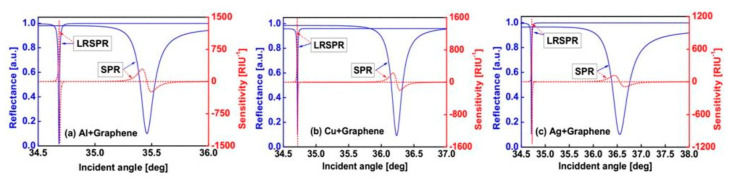
Reflectance and sensitivity on standard and long-range surface plasmon resonance (LRSPR) structure for (**a**) Al+graphene, (**b**) Cu+graphene, and (**c**) Ag+graphene (adapted with permission from Reference [37], copyright IEEE).

**Figure 6 nanomaterials-11-00216-f006:**
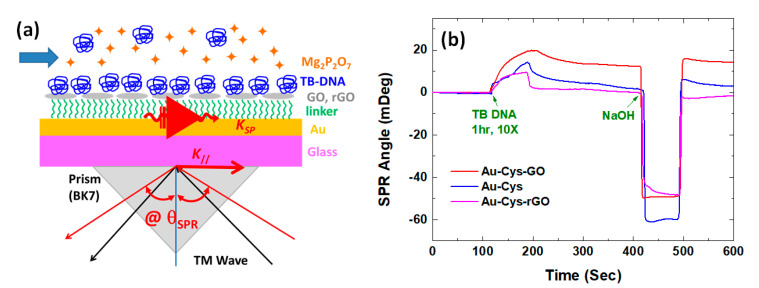
(**a**) Experimental scheme of the SPR biosensor integrated with loop-mediated isothermal amplification (LAMP) for detection of tuberculosis bacterial DNA (TB DNA). (**b**) SPR response on Cys-GO, Cys-rGO, and Cys-linker sensing surface (adapted with permission from Reference [38], copyright SPIE).

**Figure 7 nanomaterials-11-00216-f007:**
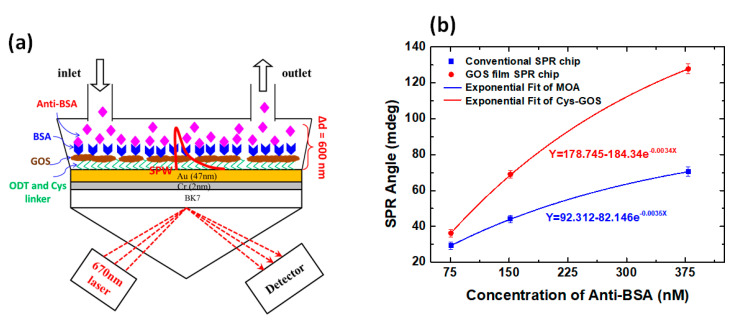
(**a**) GOS based SPR structure. (**b**) The response of the SPR biosensor at different anti-BSA concentrations (adapted with permission from Reference [40], copyright Springer Nature).

**Figure 8 nanomaterials-11-00216-f008:**
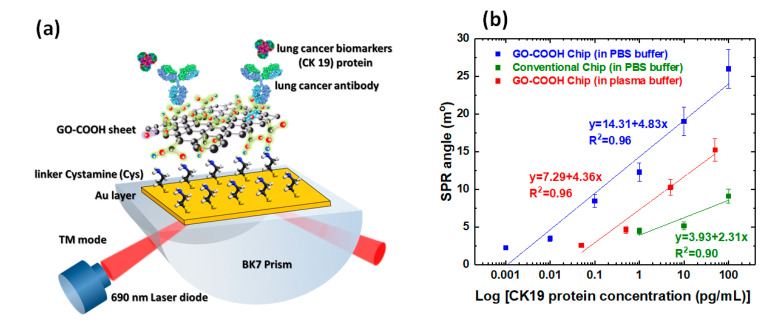
(**a**) Carboxyl-functionalized graphene oxide (GO)-based SPR structure as immunosensor to detect CK19. (**b**) Response of SPR biosensor at different CK19 concentrations and their linear ranges (adapted with permission from Reference [41], copyright Elsevier).

**Figure 9 nanomaterials-11-00216-f009:**
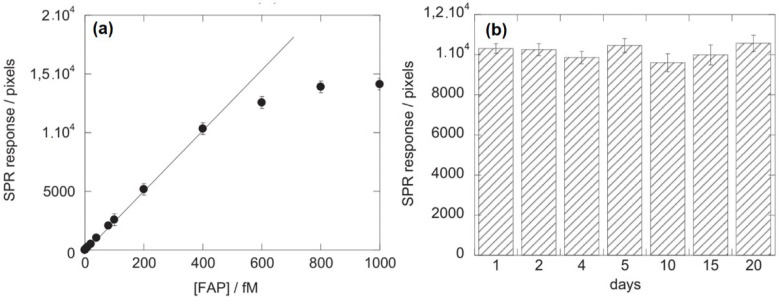
(**a**) SPR response at different FAP concentrations from 10 fM to 1 µm. (**b**) Repeatability of FAP detection for 20 days using the same sensor (adapted with permission from Reference [43], copyright Elsevier).

**Figure 10 nanomaterials-11-00216-f010:**
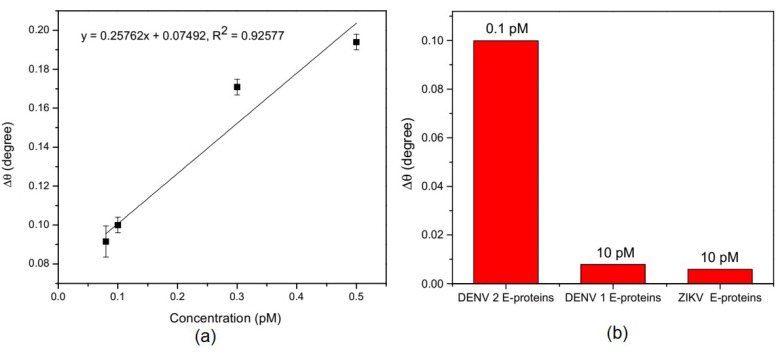
(**a**) Calibration curve or SPR angle shift at different DENV 2 E-Protein concentrations, (**b**) The response of the SPR chip to different antigens (DENV 2 E-Protein, DENV E-protein, and ZIKV E-protein) (adapted with permission from Reference [44], copyright MDPI).

**Figure 11 nanomaterials-11-00216-f011:**
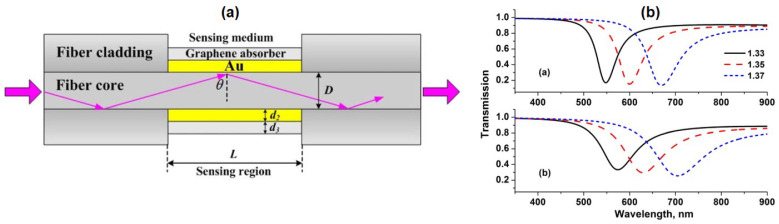
(**a**) Experimental scheme of the SPR biosensor coupled with fiber optic. (**b**) SPR spectrum at the refractive index of 1.33, 1.35, and 1.37 (top: without graphene, bottom: with graphene) (adapted with permission from Reference [57], copyright IEEE).

**Figure 12 nanomaterials-11-00216-f012:**
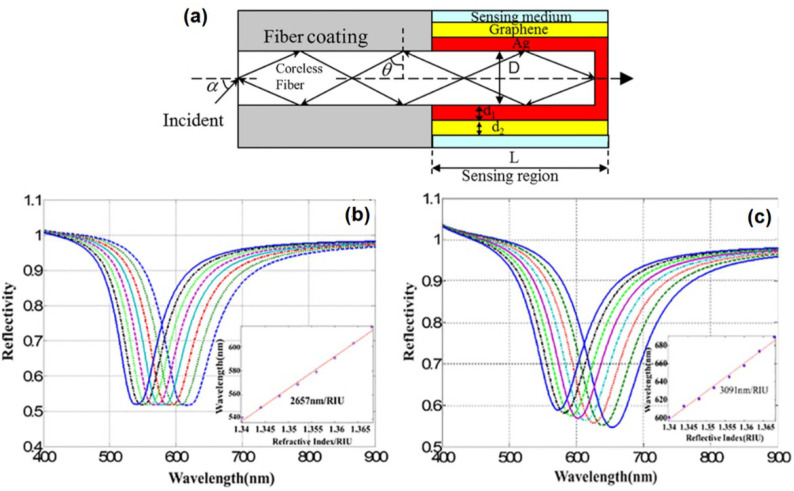
(**a**) Schematic diagram of an end reflection optical fiber with graphene. (**b**) SPR reflectance and sensitivity spectra were generated in structures without (left) and with graphene (right) (adapted with permission from Reference [58], copyright Elsevier).

**Figure 13 nanomaterials-11-00216-f013:**
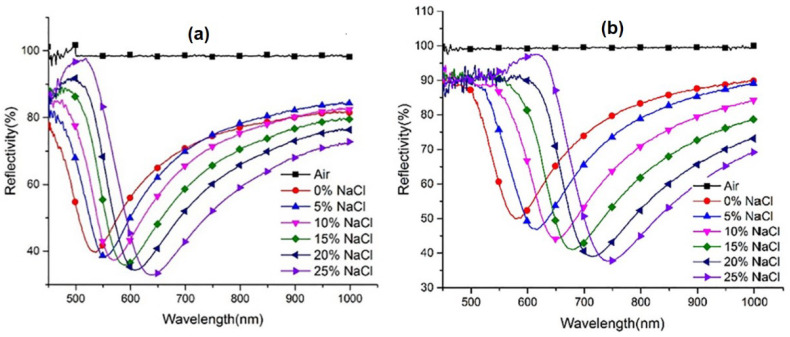
The SPR reflection spectrum at different concentrations of NaCl solution on sensing probes: (**a**) without graphene and (**b**) with graphene (adapted with permission from Reference [58], copyright Elsevier).

**Figure 14 nanomaterials-11-00216-f014:**
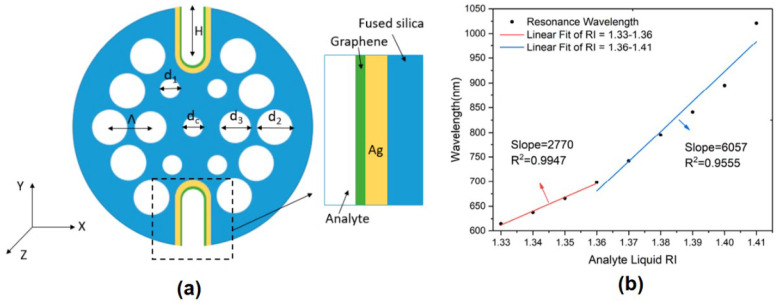
(**a**) Schematic of the designed Ag–Graphene coated photonic crystal fiber (PCF)–SPR sensor. (**b**) The relationship between the resonant wavelength and the analyte’s refractive index varies from 1.33 to 1.41 (adapted with permission from Reference [59], copyright MDPI).

**Figure 15 nanomaterials-11-00216-f015:**
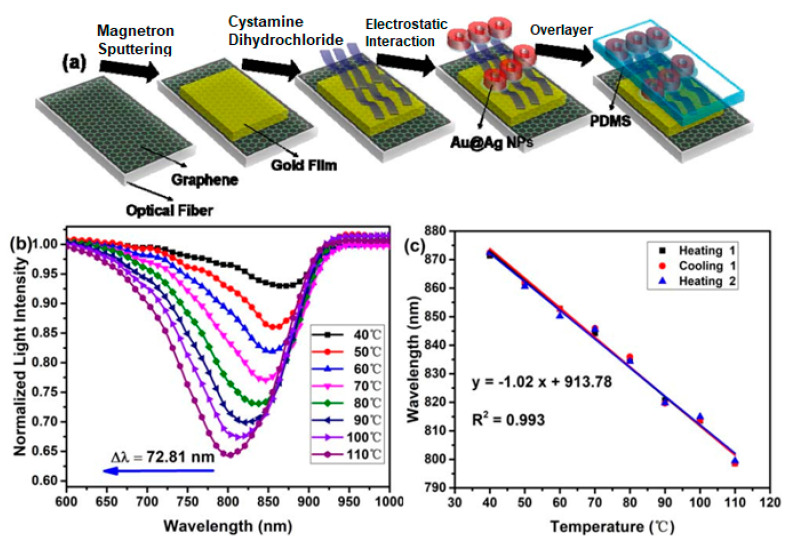
(**a**) SPR probe fabrication process. (**b**) SPR transmittance spectrum at different temperatures from 40 °C to 110 °C. (**c**) SPR response and fitting curve after heating and cooling process (adapted with permission from Reference [61], copyright IEEE).

**Figure 16 nanomaterials-11-00216-f016:**
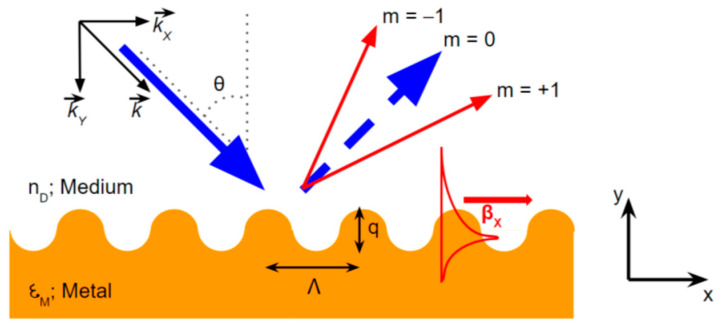
Schematic diagram of the grating-based SPR biosensor (adapted with permission from Reference [73], copyright MDPI).

**Figure 17 nanomaterials-11-00216-f017:**
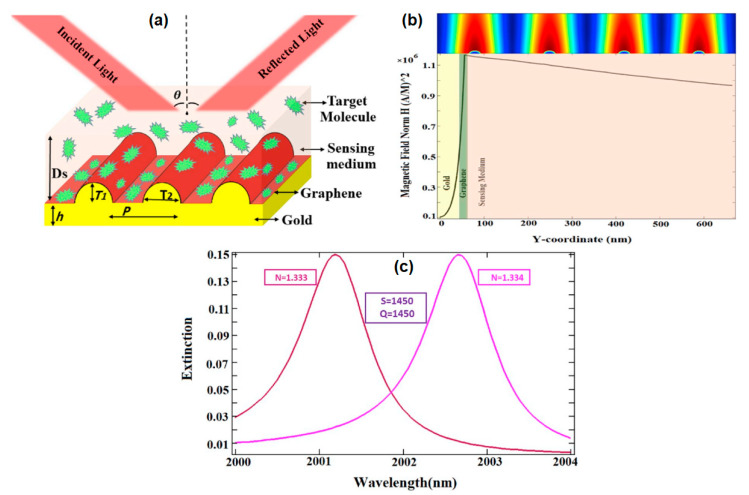
(**a**) A schematic diagram of graphene–gold grating. (**b**) The intensity distribution of the electromagnetic field at maximum wavelength of SPR mode. (**c**) SPR sensitivity investigations based on the shift in peak wavelength on the extinction curve (adapted with permission from Reference [74], copyright Elsevier).

**Figure 18 nanomaterials-11-00216-f018:**
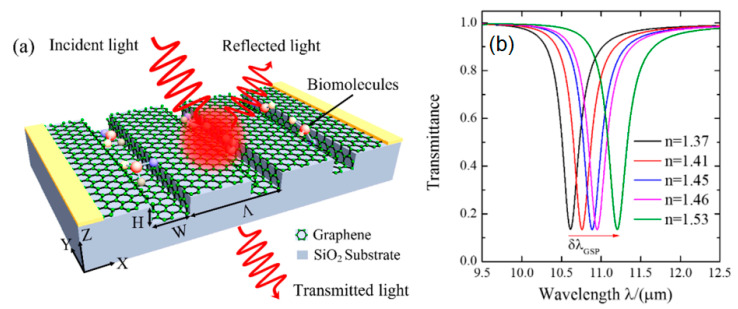
(**a**) Schematic of the conformal graphene-decorated nanofluidic channel (CGDNC) infrared sensor. (**b**) Investigation of sensor sensitivity based on data on the transmittance spectrum (adapted with permission from Reference [77], copyright MDPI).

**Figure 19 nanomaterials-11-00216-f019:**
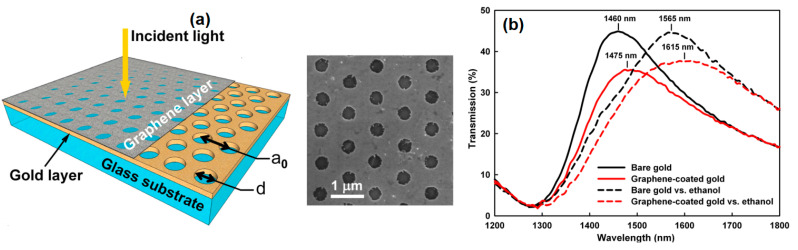
(**a**) Sketch of the developed SPR biosensor and SEM image results. (**b**) The response of the biosensor with bare gold and graphene coated gold after exposure to ethanol (Reproduced from [79], with the permission of AIP Publishing).

**Figure 20 nanomaterials-11-00216-f020:**
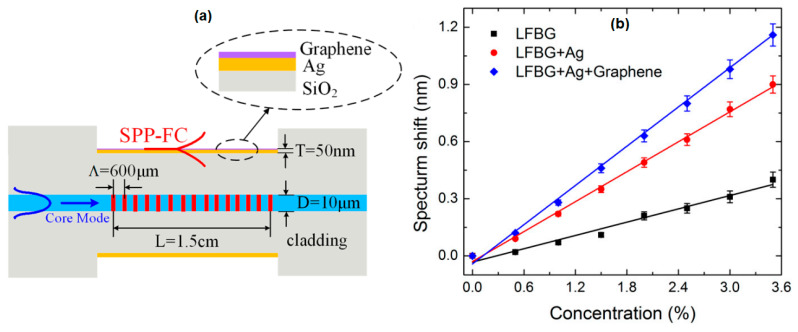
(**a**) Longitudinal section of the graphene-based long period fiber grating (LPFG) SPR sensor. (**b**) Resonance wavelength shift versus concentration of methane (adapted with permission from Reference [80], copyright MDPI).

**Figure 21 nanomaterials-11-00216-f021:**
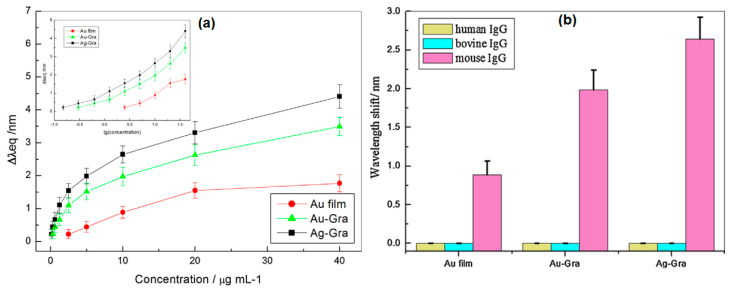
(**a**) SPR response at three different nanoparticles (Au film, Au-Gra, Ag-Gra). (**b**) The selectivity of the SPR on three different analytes (human IgG, bovine IgG, and mouse IgG) (adapted with permission from Reference [84], copyright Elsevier).

**Figure 22 nanomaterials-11-00216-f022:**
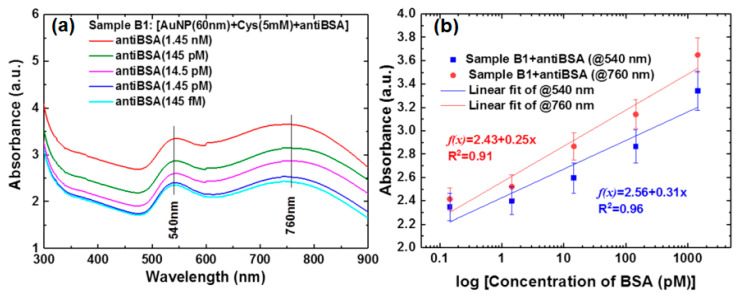
(**a**) The relationship between optical absorbance and wavelength at different anti BSA concentrations (145 fM-1.45 nM). (**b**) Calibration curves obtained from shifts in absorbance peaks (A and B) due to detection of anti-BSA with different concentrations (adapted with permission from Reference [85], copyright Springer Nature).

**Figure 23 nanomaterials-11-00216-f023:**
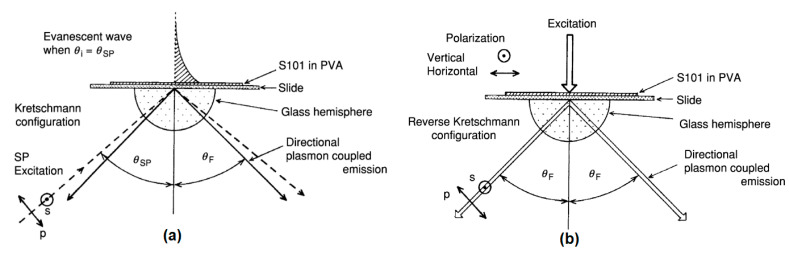
The excitation and emission of surface plasmon coupled emission (SPCE) in (**a**) Kretschmann (KR) configuration, (**b**) reverse Kretschmann (RK) configuration (adapted with permission from Reference [93], copyright Elsevier).

**Figure 24 nanomaterials-11-00216-f024:**
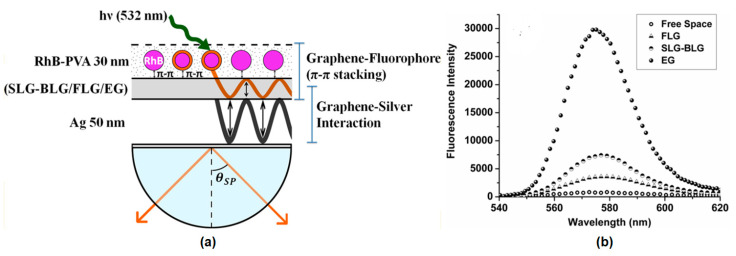
(**a**) Schematic depicting the graphene–fluorophore (π–π stacking) and graphene–silver (plasmon–plasmon coupling) interactions. (**b**) Enhancement plot displaying intensity of the SPCE of the different Ag–graphene (Single layer graphene (SLG), bilayer graphene (BLG), few layered graphene (FLG), and exfoliated graphene (EG)) versus the intensity of the free space emission (adapted with permission from Reference [94], copyright American Chemical Society).

**Figure 25 nanomaterials-11-00216-f025:**
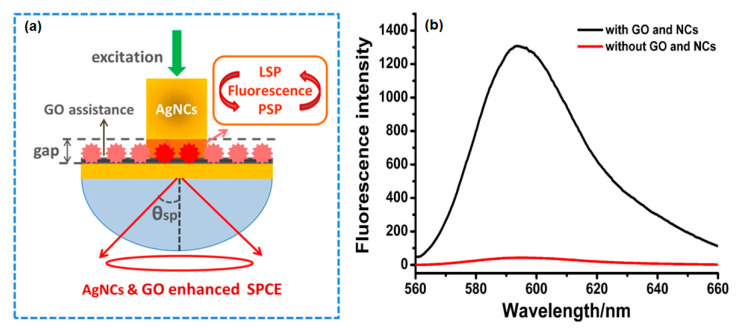
(**a**) AgNCs and GO enhanced SPCE structure. (**b**) The resulting fluorescence spectrum (right) (adapted with permission from Reference [95], copyright Elsevier).

**Figure 26 nanomaterials-11-00216-f026:**
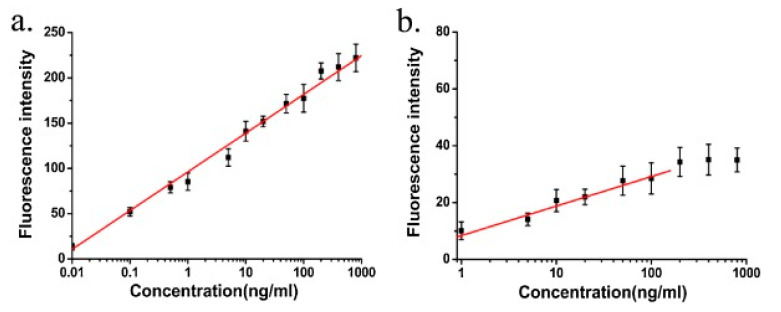
Dependences of the fluorescence intensities of (**a**) SPCE (Au+GO) and (**b**) SPCE (Au) on the concentration of human IgG (adapted with permission from Reference [96], copyright Elsevier).

**Figure 27 nanomaterials-11-00216-f027:**
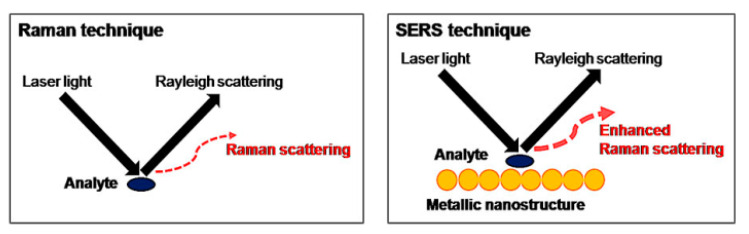
The principles behind Raman and Surface-Enhanced Raman Scattering (SERS) techniques (adapted with permission from Reference [107], copyright Institute of Food Technologists).

**Figure 28 nanomaterials-11-00216-f028:**
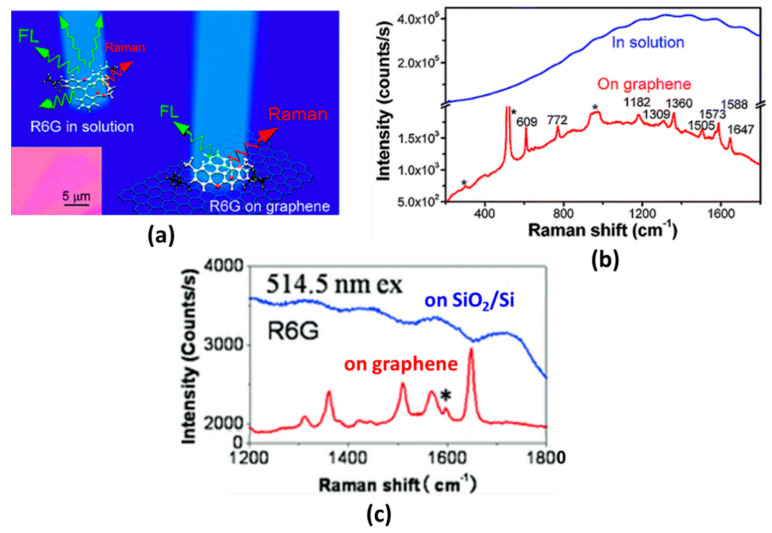
(**a**) Schematic illustration of graphene as a substrate for suppressing the photoluminescence of R6G. (**b**) Raman spectrum of R6G in water (blue line) and R6G in single layer graphene (red line). (**c**) Raman spectra of R6G on SiO_2_/Si and graphene substrate, respectively (adapted with permission from References [109,110], copyright American Chemical Society).

**Figure 29 nanomaterials-11-00216-f029:**
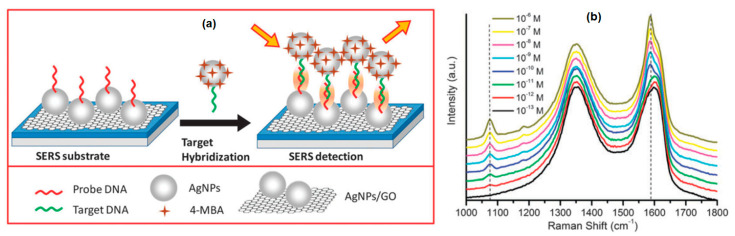
(**a**) Illustration of detection of DNA targets with Ag NPs-Ag NPs-GO. (**b**) The SERS spectrum was obtained at different complement DNA target concentrations (Reproduced from Reference [113] with permission from the PCCP Owner Societies).

**Table 1 nanomaterials-11-00216-t001:** Physical properties of graphene [15].

Physical Properties	Graphene Oxide (GO)	Reduced Graphene Oxide (rGO)	Pure Graphene
Tensile strength	~0.13 GPa	Unknown	~130 GPa
Elastic modulus	23−24 GPa	250±150 GPa	1000 GPa
Elongation at break	0.6%	unknown	0.8%
Electrical conductivity	Non conductive	~667 S/m	~1000 S/m
Dispersibility in water	Highly dispersible	Moderately dispersible	Not dispersible

**Table 2 nanomaterials-11-00216-t002:** Comparison of the sensitivity of graphene-based prism coupled SPR biosensor with relevant work based on experiment results.

SPR Structure	Target	Linear Range	Sensitivity	Ref.
BK7/Au/WSe_2_/Graphene	-	1.3–1.38	178.87°/RIU	[45]
BK7/ZnO/Au/Graphene	Bacteria	1.33–1.4	187.43°/RIU	[46]
Prism/TiO_2_/ZnO/Au/MoS_2_/GO	-	1.3–1.38	210.75°/RIU	[47]
BK7/Ag/BaTiO_3_/Graphene	-	1.338–1.353	257°/RIU	[48]
N-FK51A/Au/Graphene	Glucosa	1.338–1.405	275.15°/RIU	[49]
Au/graphene	Anticholera toxin	0.004–4 ng/mL	-	[50]
Au/graphene	Folic acid protein	5 fM	25.85/M	[43]
Au/GO/COOH	Anti-BSA	0.01 pg/mL	450.67 m/[μg/mL]	[51]
Au/GO	hCG protein	0.065–250 nM	38.34 m/nM	[52]
Au/GO/COOH	CK19 protein	0.001–100 pg/mL	4.83 m/[pg/mL]	[41]
Au/GO/COOH	Anti-PAPP-A2	0.01–10^4^ pg/mL	8.34 m/[pg/mL]	[53]
Au/graphene	Glycerol	1.33–1.36	179.79°/RIU	[54]

**Table 3 nanomaterials-11-00216-t003:** A summary of selected papers regarding graphene-based optical fiber coupled SPR biosensor.

Fiber optic Type	Structure	Target	Sensitivity and Limit of Detection (LOD)	Ref.
Side polished optical fiber	Au/graphene	ssDNA	1039.8 nm/RIU and 10^−12^ M	[65]
Plastic clad silica fiber	Au/graphene	BSA	6500 nm/RIU	[66]
Plastic clad silica fiber	Au/graphene	BSA	7.01 nm/(mg/mL)	[66]
End reflection optical fiber	Ag/graphene	NaCl solutions	3936.8 nm/RIU	[58]
U-bent plastic optical fiber	Graphene + Ag nanoparticles	Glucosa solutions	700.3 nm/RIU	[67]
MMF-PCF-MMF sensor	Au+graphene+SPA	anti-human IgG	4649.8 nm/RIU	[68]
D-shaped fiber	Cr/Au/MoS2/graphene	Glucosa	6708.87 nm/RIU	[69]

Note: SPA: The staphylococcal protein A.

**Table 4 nanomaterials-11-00216-t004:** A summary of selected papers regarding graphene-based grating coupled SPR biosensor.

Grating shape	Structure	Analyte	Sensitivity	Ref.
Rectangular	Au/graphene	dangerous gases	1180 nm/RIU	[75]
Ellipse	Au/graphene	Biological cells	1782 nm/RIU	[74]
Rectangular	SiO_2_/graphene	ssDNA	8004 nm/RIU	[77]
Holey	Au/graphene	Ethanol	-	[79]
Long period fiber grating (LPFG)	Ag/graphene	Methane	0.344 nm/%	[80]
Rectangular	Ag grating/Ag/graphene	-	220°/RIU	[81]

**Table 5 nanomaterials-11-00216-t005:** A summary of selected papers regarding graphene-based nanoparticle coupled SPR biosensor.

Technique	Structure	Analyte	Sensitivity	LOD	Ref.
SPR	two layers of GO-Au NPs composite	miRNA-141	-	0.1 fM	[87]
LSPR	Au NPs-GO-anti BSA	hCG	-	145 fM	[85]
SPR	Au-(Au NPs-Graphene nanohybrids)-anti human IgG	mouse IgG		0.15 µg/mL	[84]
LSPR	Au NPs coupled with GO	NO_2_	-	-	[86]
SPR	Graphene-coated SPR with Au nanostars carrying ssDNA	ssDNA	-	500 aM	[88]
LSPR	Au NPs/GO/uricase	Uric acid	0.0082 nm/µM	206 µM	[89]

**Table 6 nanomaterials-11-00216-t006:** A summary of selected papers regarding graphene-based plasmon coupled emission biosensor.

Structure	Enhancement Factor	Fluorophore	Ref.
Ag/graphene	40	RhB–PVA	[94]
Au/GO	25	RhB–PVA	[96]
Ag/single layer GO	112	R6G	[97]
Au/Ag NCs/GO	30	AgNCs	[95]
Ag/Gallium arsenide/Ag/graphene/MoS_2_	4.772	-	[98]

**Table 7 nanomaterials-11-00216-t007:** A summary of selected papers regarding the graphene-based Surface-Enhanced Raman Scattering (SERS) biosensor.

SERS Platform	Probe Molecule	Enhancement Factors (EF)	LOD	Ref.
Graphene-encapsulated Ag NPs decorated silicon nanowire	R6G	10^7^	-	[124]
GO-Au NPs composites	R6G	4.9 × 10^6^	10^−9^ M	[125]
GO-based Au hybrids	R6G	1.2 × 10^7^	10^−7^ M	[126]
Graphene-Au nano-pyramid (tip) hybrid structure	R6G	-	-	[127]
Au NPs arranged on GO	R6G	-	10^−9^ M	[106]
Self assembly of Ag NPs into Ag NPs-GO nanocomposites	R6G	-	10^−14^ M	[113]
Au NPs/graphene/epoxy resin nanosheet	-	6.2 × 10^6^	3.3 μM	[128]
Graphene functionalized with polyamidoamine dendrimers decorated Ag NPs	-	8.3 × 10^4^	1.43 pM	[129]

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
