# Peer review of "A Review of Graphene-Based Surface Plasmon Resonance and Surface-Enhanced Raman Scattering Biosensors: Current Status and Future Prospects"

_nanomaterials, 2021, doi:10.3390/nano11010216_

Round 1

Reviewer 1 Report

This is a timely review about graphene-based SPR detection. The manuscript is overall well organized. I have only minor comments about references.

  1. The title is somewhat confusing by having Future in it, because most of what the manuscript address is either the past or the current status of the technology. Perhaps a better one that the authors can start with may be something like “A Review on Graphene-Based Surface Plasmon Resonance (SPR) Bio-sensors: Current Status and Future Prospects”.
  2. Graphene-based SERS is related to plasmon excitation, but is not SPR in itself. In this sense, authors may also want to extend the title to include SERS.
  3. 29 & 30 are redundant. So are Refs. 31 & 32. Please check more thoroughly for any more redundancy.
  4. Despite being a full review, many works that may be relevant are left uncited and I encourage authors to cite more related studies in this regard, e.g., - Chung, et al. "Enhancing the Performance of Surface Plasmon Resonance Biosensor via Modulation of Electron Density at the Graphene–Gold Interface." Adv. Mater. Interfaces 5.19 (2018): 1800433 / Chung, et al. "Systematic study on the sensitivity enhancement in graphene plasmonic sensors based on layer-by-layer self-assembled graphene oxide multilayers and their reduced analogues." ACS Appl. Mater. Interfaces 7.1 (2015): 144-151 / Ryu, et al. "Effect of coupled graphene oxide on the sensitivity of surface plasmon resonance detection." Appl. Opt. 53.7 (2014): 1419-1426.
  5. Please check consistency of abbreviations and usage of English throughout the manuscript.

Reviewer 2 Report

This paper discusses the current development of a graphene-based SPR biosensor for various excitation methods. The discussion begins with a discussion regarding the properties of graphene in general and its use in biosensors. Simulation and experimental results for several excitation methods are then presented to provide an overview of the development of materials in the biosensor in the future.

This is indeed an interesting work, the authors have covered the topic, they present adequate references, and can be published after minor revision.

A few typos, and some syntax issues should be resolved.

I suggest the authors to add information regarding the use of graphene based metamaterials for bio-sensors. This is also a topic that they should cover.
